# Current and Future Biomarkers in Multiple Sclerosis

**DOI:** 10.3390/ijms23115877

**Published:** 2022-05-24

**Authors:** Jennifer Yang, Maysa Hamade, Qi Wu, Qin Wang, Robert Axtell, Shailendra Giri, Yang Mao-Draayer

**Affiliations:** 1Department of Neurology, Clinical Autoimmunity Center of Excellence, University of Michigan Medical School, Ann Arbor, MI 48109, USA; yangjen@umich.edu (J.Y.); hamadem@med.umich.edu (M.H.); qiw@med.umich.edu (Q.W.); qinwang@med.umich.edu (Q.W.); 2Department of Arthritis and Clinical Immunology Research, Oklahoma Medical Research Foundation, Oklahoma City, OK 73104, USA; bob-axtell@omrf.org; 3Department of Neurology, Henry Ford Health System, Detroit, MI 48202, USA; sgiri1@hfhs.org; 4Graduate Program in Immunology, Program in Biomedical Sciences, University of Michigan Medical School, Ann Arbor, MI 48109, USA

**Keywords:** multiple sclerosis, biomarkers, neurofilament light chain, cytokines, metabolites, sCD40L, microbiome, prognosis, disease progression

## Abstract

Multiple sclerosis (MS) is a debilitating autoimmune disorder. Currently, there is a lack of effective treatment for the progressive form of MS, partly due to insensitive readout for neurodegeneration. The recent development of sensitive assays for neurofilament light chain (NfL) has made it a potential new biomarker in predicting MS disease activity and progression, providing an additional readout in clinical trials. However, NfL is elevated in other neurodegenerative disorders besides MS, and, furthermore, it is also confounded by age, body mass index (BMI), and blood volume. Additionally, there is considerable overlap in the range of serum NfL (sNfL) levels compared to healthy controls. These confounders demonstrate the limitations of using solely NfL as a marker to monitor disease activity in MS patients. Other blood and cerebrospinal fluid (CSF) biomarkers of axonal damage, neuronal damage, glial dysfunction, demyelination, and inflammation have been studied as actionable biomarkers for MS and have provided insight into the pathology underlying the disease process of MS. However, these other biomarkers may be plagued with similar issues as NfL. Using biomarkers of a bioinformatic approach that includes cellular studies, micro-RNAs (miRNAs), extracellular vesicles (EVs), metabolomics, metabolites and the microbiome may prove to be useful in developing a more comprehensive panel that addresses the limitations of using a single biomarker. Therefore, more research with recent technological and statistical approaches is needed to identify novel and useful diagnostic and prognostic biomarker tools in MS.

## 1. Introduction

Multiple sclerosis (MS) is a chronic, immune-mediated inflammatory demyelinating disease of the central nervous system (CNS) that affects more than 2 million people around the world, with the highest prevalence among those between the ages of 20 and 40 years [1,2]. There are currently many treatment options to reduce the relapse rate and neuroinflammation in relapsing-remitting multiple sclerosis (RRMS), the most common clinical subtype of MS; however, after 10–20 years, most patients with RRMS progress to secondary progressive multiple sclerosis (SPMS) with eventual axonal and neuronal degeneration [3]. Primary progressive multiple sclerosis (PPMS) is another progressive form of MS, and along with SPMS, there are few effective treatments that can prevent the disability progression in these two clinical subtypes. Ocrelizumab and Siponimod, though approved for progressive disease, are only marginally effective in SPMS and PPMS patients and are more effective in progressive patients with inflammatory disease activity.

Unlike rheumatoid factor (RF) and cyclic citrullinated peptide (CCP) used to diagnose rheumatoid arthritis (RA), there is no serologic diagnostic test for MS [4]. When a patient exhibits symptoms classic of MS, they are tested to exclude other diseases, such as lupus, Lyme disease, and B12 deficiency. It is also sometimes difficult to determine whether a patient with MS is experiencing a true relapse or a pseudo-relapse confounded by infection and comorbidities [5]. Not only are biomarkers for MS diagnosis and true relapses lacking, but biomarkers for disease progression and treatment response are also largely lacking. The latter has been hindered by the heterogeneity in the immune signatures in the patient population, affecting the development of effective treatment for progressive disease [6,7].

Although it is effective to use the annualized relapse rate to determine treatment efficacy in RRMS, due to the insensitivity of the Expanded Disability Status Scale (EDSS), it is much more challenging to quantify disability progression in progressive MS. The Multiple Sclerosis Functional Composite (MSFC), 25-foot walk test (T25FW), 9-hole peg test (9-HPT), and the symbol digit modality tests (SDMT) have been incorporated in trials to increase the sensitivity [8]. It is crucial to find sensitive biomarkers that can predict disease progression. An ideal biomarker should have diagnostic and prognostic value, correlate with specific disease activity such as relapse or progression, respond to treatment, and essentially be useful in clinical trial outcomes. Additionally, an ideal biomarker should be non-invasive, safe, accurate, reproducible, cost-effective, and easily detectable in patients [9].

## 2. Classic Diagnostic Markers

### 2.1. Magnetic Resonance Imaging

Currently, the most reliable and routinely used diagnostic tool for MS is magnetic resonance imaging (MRI). Specifically, T2-weighted MRI images are used to identify both white matter and gray matter MS lesions. The presence of these lesions demonstrates inflammation with mixed pathology of neuro-axonal damage and demyelination [10]. However, confirming relapses and active inflammation in RRMS is done by detecting gadolinium-enhancing T1 lesions on MRIs. The presence of enhancing brain lesions has also been associated with long-term disability and brain atrophy, with new or enlarging lesions being correlated with disability progression and cerebral atrophy. However, routine MRIs do not specifically detect damage to neurons and axons, which is most strongly correlated with long-term disability in MS [3,11,12]. Routine MRIs lack sensitivity and specificity for neurodegeneration; measuring cortical thickness at an individual level may be challenging with interpersonal variability [13].

### 2.2. Spinal Fluid Analysis

The main advantage of using cerebrospinal fluid (CSF) over blood to measure biomarkers is that it more accurately reflects the inflammatory profile of the CNS [14]. CSF biomarkers have the advantage of being more sensitive compared to clinical or MRI assessments, especially in the setting of low-grade disease activity in MS. Among some patients whose disease was considered inactive by clinical scales and/or MRI, CSF neurofilament light chain (cNfL) and immunoglobulin (Ig)G-index were found to be significantly elevated [15]. IgG index is the ratio of IgG to albumin in the CSF compared to that in the serum [16]. Having a ratio greater than 0.7 usually results in a diagnosis of MS. Analyzing the CSF for markers of inflammation such as oligoclonal bands and IgG index is helpful in diagnosis, but oligoclonal bands and IgG index are not ideal biomarkers for predicting relapse and progression. Measuring oligoclonal bands is not very sensitive as it can be challenging to determine how many bands are present. Additionally, they are not very specific, as anything causing chronic inflammation can result in elevated oligoclonal bands [17]. However, some studies have shown that IgM-type oligoclonal bands are associated with increased MS activity, increased retinal axonal loss, decreased retinal nerve fiber layer, and more aggressive disease progression during early stages of RRMS [16,18,19,20,21].

### 2.3. Evoked Potentials

Evoked potentials (EPs), including visual evoked potentials (VEPs), somatosensory evoked potentials (SSEPs), and brainstem auditory evoked response (BAERs), are non-invasive techniques that can assess neural conduction in various pathways. It is performed by stimulating the respective system, with a scalp electrode over the corresponding cortex measuring the latency and amplitude. Prolonged latency indicates damage due to demyelination, which can be a useful tool and aid in diagnosing MS and assessing specific pathways [22]. There is also a proposed role of EPs in the prognosis of MS and the treatment response, but this is not yet widely clinically applicable [23].

## 3. Other Imaging Techniques

### 3.1. Optical Coherence Tomography

Optical coherence tomography (OCT) is a non-invasive assessment that uses light to scan the retina and optic disc to measure the degeneration of the optic nerve after inflammation. This has been proposed as a model to examine neurodegeneration. Measures have been incorporated into clinical trials for neuroprotective agents. The thickness of the retinal nerve fiber layer (RNFL) directly measures optic nerve axonal loss, which is associated with a persistent visual deficit. RNFL loss is associated with brain atrophy, neurological impairment, and disease duration [24].

### 3.2. Magnetic Transfer Imaging

Magnetic Transfer Imaging (MTI) can help provide information on the severity of the disease by differentiating between MS lesions and tracking the evolution of acute lesions. This is done by calculating the magnetic transfer ratio (MTR), which provides information about tissue integrity by its ability to exchange magnetization with mobile water molecules. This can elucidate the level of intact white matter, which has been shown to be associated with the extent of demyelination and axonal loss [25,26,27]. MTR can also be calculated for the optic nerve, with a lower value indicating decreased RNFL and axonal degeneration [28].

### 3.3. Magnetic Resonance Spectroscopy

Magnetic Resonance Spectroscopy (MRS) can be used to determine the extent of CNS cellular metabolism via a non-invasive technique measuring several biochemical molecules. It measures N-acetylaspartate levels, which have been implicated in neuronal and axonal loss. MS lesions around normal-appearing white matter and cortical grey matter have shown lower levels of N-acetylaspartate (NAA) [29], which has been associated with disease progression and disability [30]. Additionally, choline is measured to determine whether there has been an increase in rotational cell membrane components, which is usually seen with demyelination or gliosis. It also measures glutamate levels, which correlate with acute inflammation, and γ-aminobutyric acid (GABA), which has been found to be decreased in SPMS [31].

### 3.4. Diffusion Weighted Imaging

Diffusion weighted imaging (DWI) helps to differentiate between various CNS pathologies, such as infections, strokes, tumors, and neurodegeneration. This is done by evaluating water’s apparent diffusion coefficient in the brain, which is associated with changes in cell structure and white matter tracts. Specifically, it is helpful in evaluating ischemic cerebrovascular accidents and is limited in its role in diagnosing MS as it is hindered by its inability in measuring extent of tissue loss in MS lesions [32,33]. Additionally, this form of imaging has a high risk of false positives [34].

### 3.5. Diffusion Tensor Imaging

Diffusion tensor imaging (DTI) can elucidate more information about MS pathogenesis than current MRI techniques. It can measure water’s three-dimensional diffusion and provide information on axial diffusivity (DA), radial diffusivity (RD), mean diffusivity (MD), and fractional anisotropy (AF) [33]. DA can be used to examine axonal loss and degeneration, with a higher value seen with later MS disease progression. RD can be used to examine the extent of demyelination [35,36,37,38]. MD is associated with tissue changes that occur because of the breakdown of the blood-brain barrier and injuries that occur after the restoration of the blood-brain barrier (BBB), allowing it to be a useful parameter in assessing for the MS onset and relapse [39]. Failure analysis can be performed to determine whether there is an increase in global water diffusion in white matter tracts and a decrease in disorganized fibers due to MS pathology [40].

## 4. Biomarkers of Axonal Damage

### 4.1. Neurofilament Light Chain

Over the past three decades, many assays have been developed to detect neurofilament light chain (NfL) levels. Neurofilaments are cytoskeletal proteins released from damaged axons into the CSF and the blood. Studies have also found that increased cNfL levels are correlated with increased CD4+ T lymphocytes, which have been implicated in the inflammation seen in MS [41], and progression of RRMS to SPMS [42]. Early studies have found that cNfL levels of MS patients are increased during active relapse and acute relapse compared to healthy controls [43]. There is a positive correlation between cNfL and serum NfL (sNfL) in MS patients [44,45,46], with cNFL levels 42-fold higher than sNFL levels [45]. A benefit to using sNfL levels as opposed to cNfL levels is that serum levels are easier to obtain than conducting a spinal tap or lumbar puncture on a patient to retrieve CSF. Over the last few years, single molecular array (SiMoA) has made measuring NfL concentration levels more clinically relevant.

In general, MS patients also had higher sNfL levels before treatment compared to healthy controls. With disease-modifying treatment, sNfL levels were lower [45]. Treatments with higher efficacies have also been shown to reduce NfL levels more effectively than traditional treatment options. Additionally, levels of sNfL have been associated with T2 lesion volumes [47]. Some reports showed strong correlations between sNFL levels and the number of active lesions present on MRI scans [48,49]. However, some patients have several active MRI lesions with low sNfL, and some have no MRI lesions with high sNfL, indicating that other confounding factors can result in high sNfL levels. Therefore, patients will still require MRI scans [49]. Studies have also shown that brain and spinal cord atrophy may be positively correlated with sNfL levels. One study showed a reduction in the brain and spinal cord volume over five years, with a greater reduction for those with higher sNfL baseline levels [50].

A recent study assessed the prognostic value of sNfL obtained close to the time of MS onset with long-term clinical outcomes [51]. sNfL were tested from samples collected at the time of diagnostic workup. After 15 or more years of follow-up, the baseline median sNfL at higher levels had a significantly higher hazard ratio of developing an EDSS ≥ 4. There was a trend toward a higher median sNfL level in patients with progressive disease, but this did not reach significance. Another study showed no association between higher long-term EDSS scores and higher sNfL levels at disease onset nor an association between sNfL levels and relapse activity overtime in MS patients [47].

A study of United States (U.S.) military personnel with MS and their matched controls assessed whether sNfL levels are elevated before the clinical onset of MS [52]. The sNfL levels were higher in cases of MS patients compared to controls. In the group with two pre-symptomatic samples, the levels were higher in MS patients closer to the onset of symptoms (median increase of 1.3 pg/mL per year), with no significant difference in the samples from the matched controls over time. In the second group where samples were collected before and after symptom onset, most had a significant increase in sNfL levels between the two points (median level of 25.0–45.1 pg/mL), suggesting that neuroaxonal degradation starts years prior to the onset of MS and emphasizing the importance of early treatment. A follow-up study further demonstrated that in a cohort of U.S. military personnel, an infection of Epstein-Barr Virus (EBV) increased the risk of MS by 32-fold. However, this was not observed for other viral infections. sNFL levels were also found to be elevated after EBV seroconversion [53].

The sNfL level is a valuable biomarker at the group level. However, it may be challenging to use in a clinical setting to assess whether an individual has MS. Many studies showed that there is a large overlap between the baseline sNfL level of patients with MS and their controls who may have migraine or conversion disorder [47]. Using NfL levels as a biomarker for MS relapse is not specific, as NfL levels are elevated in infections and many neurodegenerative [54] and neurological disorders in addition to MS. sNfL is positively correlated with age due to age-related neuronal degeneration and has been found to be higher in older patients during relapse [45]. This may be a significant confounding variable because progressive MS patients tend to be older patients. Plasma NfL (pNfL) levels are also negatively correlated with body mass index (BMI) and blood volume [55]. Therefore, the search for a predictive and diagnostic biomarker for MS continues because sNfL cannot, itself, be individually used to determine MS disease activity. Another application of sNfL is to monitor inflammatory disease activity and distinguish actual relapses from pseudo-relapses. Several studies have shown that sNfL levels are elevated in relapse in an MS population. However, there is a significant overlap in sNfL levels in RRMS patients, highlighting the limitations of sNfL in identifying relapses [56].

### 4.2. Tau Protein

Tau protein, which has been implicated in Alzheimer’s disease, is responsible for stabilizing axonal microtubules and has been found to be released upon neuronal damage, allowing it to be measured in the CSF [57,58]. In MS, it can be used as a biomarker for axonal loss [30]. One study found tau protein levels to be correlated with the severity of clinical symptoms [59], while another found that those with higher CSF tau protein levels tend to have a quicker disease progression measured by a one-point increase in EDSS score and can be used to predict the time to next relapse [60]. However, one study with patients with clinically isolated syndrome (CIS), which is defined as the first neurological episode of inflammation or demyelination in at least one site in the CNS lasting greater than 24 h [61], did not find a statistically significant difference in tau concentration compared to the control group nor a statistically significant correlation between tau concentration and EDSS scores [62]. However, another study found that tau protein was correlated with EDSS in both CIS and RRMS patients and that higher levels of tau were correlated with the conversion of CIS into clinically definite MS. They also found that tau protein level was associated with the number of T2-lesions on MRI [63]. Another study also found tau protein concentrations in the CSF of MS patients of all clinical subtypes to be similar to those of controls [64].

### 4.3. Amyloid-Precursor Protein

Amyloid precursor protein (APP) has also been implicated in Alzheimer’s disease but nevertheless may be associated with MS. It is produced by astrocyte cells during demyelination and can be located in reactive glial cells during de- and remyelination [65]. MS patients present with higher levels of APP compared to controls and axons that are positive for APP in MS patients have been shown to be correlated with CNS lesion development [66].

### 4.4. Tubulin Beta

Tubulin *beta* (TUB*β*) is a subunit of tubulins, which are heterodimeric proteins that make up microtubules. The neuron development and regeneration have been associated with increased production of the class II tubulin isotype. Specifically, one study found that CSF TUB*β* was increased in patients with MS compared to patients with other neurological diseases [67]. A summary of this biomarker and the ones discussed above in this section can be found in Table 1.

## 5. Biomarkers of Neuronal Damage

### 5.1. 14-3-3 Protein

14-3-3 protein, which is present in neurons, can be measured in the CSF of both patients with MS and those with Creutzfeldt-Jakob disease [64]. However, the role of 14-3-3 protein in MS is inconsistent. Studies have found that 14-3-3 protein in the CSF is associated with more severe disability, more extensive involvement of the spinal cord, and quicker progression to MS or disease progression [68,69,70]. Early accumulation of 14-3-3 protein in the CSF may be correlated with decreased rates of recovery. However, some studies have difficulties detecting 14-3-3 protein in the CSF, with one only detecting it in 2 patients out of 22 MS patients [71] and another detecting it in only one patient out of 21 CIS patients [62].

### 5.2. Neuron Specific Enolase

Concentrations of neuron specific enolase (NSE), an enzyme found in neurons and axons that can be used to estimate neuronal density, have been found to be increased in both the CSF and serum of patients suffering from trauma, hypoxic brain injury, or cerebral bleeding [72,73,74]. One study found a decrease in serum and CSF NSE in patients with CIS compared to healthy controls [62], with some studies showing either no change [43,74] or a negative correlation between plasma NSE levels and EDSS and Multiple Sclerosis Severity Score (MSSS) [75]. A summary of this biomarker and the ones discussed above in this section can be found in Table 2.

## 6. Biomarkers of Glial Dysfunction

### 6.1. Glial Fibrillary Acidic Protein

Glial fibrillary acidic protein (GFAP) is expressed by mature astrocytes and has been found to be increased in plaques of MS patients, indicating damage to astrocytes [76,77]. Patients with SPMS had higher CSF GFAP levels than those with RRMS [78]. Additionally, higher CSF levels of GFAP are associated with greater disabilities and relapse [79].

### 6.2. S100β Protein

There have been reports of an increase in serum and plasma levels of S100*β* protein, a subunit of the S100 protein that is found in glial cells, in MS [80], with the highest levels in patients with PPMS or SPMS [79]. Examples of S100*β* functions include maintaining astrocyte integrity, assisting with neuronal proliferation, and differentiating oligodendrocytes. During acute exacerbations in patients with RRMS, S100*β* levels have also been shown to be increased; however, the window is small as S100*β* levels were no longer increased in patients who had acute exacerbations prior to one week ago [80]. Changes in S100*β* have also been seen in patients with cerebral ischemia of amyotrophic lateral sclerosis [81,82]. However, a study did not find a statistically significant difference in CSF and serum S100β protein concentration between CIS patients and healthy controls, but this may be confounded by some samples being obtained more than one week after the acute exacerbation. The same study also did not find any significant correlation between S100*β* protein concentration and EDSS score [62]. No differences in plasma S100*β* protein concentration between various clinical subtypes of MS have also been described [75], along with no difference in CSF S100*β* concentration between patients with MS and controls [43].

### 6.3. Anti-Aquaporin 4 Antibodies

Astrocytes express aquaporin-4 (AQP4) to help establish homeostasis in the CNS by moving water through cell membranes. However, studies have found that AQP4 is undetectable in patients with MS. This is a measure to help with the difficult task of differentiating MS from neuromyelitis optica (NMO), which is a rare condition that also presents with demyelination of the optic nerve and spinal cord [83,84].

### 6.4. Nitric Oxide

Nitric oxide (NO) has been found to be increased in both the serum and CSF of MS patients [85,86]. It inhibits cytochrome C oxidase, resulting in impaired mitochondrial function due to decreased energy production [87]. Byproducts of NO degradation can destroy mitochondria, leading to prominent damage in MS lesions. It can also increase the effects of apoptosis on neurons and glial cells and allow the passage of pro-inflammatory cells into the CNS by increasing blood-brain barrier permeability [88]. A summary of this biomarker and the ones discussed above in this section can be found in Table 3.

## 7. Biomarkers of Myelin Biology/Demyelination

### 7.1. Myelin Basic Protein

Myelin basic protein (MBP) is produced by the oligodendrocytes from the central nervous system and has been found to be increased in the CSF of patients with MS and correlated with EDSS scores [89]. One study found that MS patients with an acute exacerbation had higher levels than those with slower progressive MS and even higher than those in remission [90]. However, using MBP is challenging, as demyelination lesions can be remyelinated by MBPs in the CSF [91,92].

### 7.2. Myelin Oligodendrocyte Glycoprotein

Myelin oligodendrocyte glycoprotein (MOG) associated disease, a newly recognized disorder, is differentiated from NMO and MS by the presence of MOG antibodies in the serum. Additionally, differences have been found in CSF myeloid cell types in subjects with neuroinflammation, which included those with MS and anti-MOG disorder [93]. A summary of this biomarker and the ones discussed above in this section can be found in Table 4.

## 8. Biomarkers of Immunomodulation and Inflammation

### 8.1. Immune Mediators and Cytokines

Pro-inflammatory cells, T helper (Th) 1 and Th17 cells, produce cytokines, such as interleukin (IL)-17, interferon (IFN)-γ, and tumor necrosis factor (TNF)-α, while anti-inflammatory cells, regulatory T (Treg) and Th2 cells, produce IL-10 and IL-4. Measuring these cytokines and cellular changes can reflect the disease type, demonstrated by a study in pediatric MS where serum levels of anti-inflammatory cytokine IL-10 were predictive of relapse compared to other pro- and anti-inflammatory cytokines [94]. Therefore, immune signatures, along with the above markers all together as a composite, can be used to further differentiate underlying disease pathology and disease activity (Figure 1).

Additionally, C-X-C motif chemokine ligand (CXCL) 13 has been found to be correlated with worse prognosis and exacerbations in RRMS and conversion of CIS to MS. However, CXCL13 is non-specific as patients with infections also had high levels [95]. One study found that CSF and plasma levels of eotaxin-1 (CCL11) were associated with disease duration, especially in patients with SPMS. They also found C-C motif chemokine ligand (CCL) 20 to be associated with disease severity and CSF levels of IL-12B, macrophage inflammatory protein (MIP)-1a, cluster of differentiation (CD)5, and CXCL9, and plasma levels of oncostatin (OSM) and hepatocyte growth factor (HGF) to be associated with MS [96]. Serum IL-6 has also been found to be correlated with the age of onset for MS patients and was detected at a higher rate in MS patients compared to controls [97].

Immune signatures may also predict treatment response or prognosis of MS patients. Most biomarker studies have focused on IFN-β, where there is a wide variation in response to this therapy. Neutralizing antibodies (Nabs) against IFN-β are associated with treatment failure [98], but they only partially explain non-responsiveness. Serum cytokine profiles have shown immunologically distinct subgroups of MS, and these subgroups may stratify treatment response to IFN-β [6].

### 8.2. Soluble CD40L

We took a novel approach by using age and disease duration-matched non-progressive benign multiple sclerosis (BMS) patients to look for progression-specific biomarkers using Luminex array. We found plasma soluble CD40L (sCD40L) was significantly increased in SPMS compared to non-progressive benign MS (BMS). While the combination of sCD40L and monocyte chemoattractant protein 1 (MCP1)/CCL2 could be used to distinguish RRMS from SPMS, elevated sCD40L and IFN-γ levels are best at differentiating SPMS from BMS [99]. We further demonstrated in a Phase I proof-of-concept study that anti-CD40L monoclonal antibody (mAb) IDEC-131 (Toralizumab) was safe and feasible for treating MS [100]. Immunological analysis showed no depletion of lymphocyte subsets. Instead, an increase in CD25+/CD3+ and CD25+/CD4+ ratio and a shift towards an anti-inflammatory cytokine response were seen. In light of the fact that sCD40L was found to be highly upregulated in SPMS compared to BMS, it further strengthens the importance of this therapeutic target. A Phase II trial in MS using a next-generation anti-CD40L mAb, SAR441344, will provide further insight into the role of the heightened level of sCD40L in SPMS patients. We also showed that RNFL thickness deteriorates only mildly in BMS and that T cells have upregulated IL-10 and leukemia inhibitory factor (LIF) and downregulated IL-6 and neurotensin high affinity receptor 1 in BMS [101]. Therefore, comparison with non-progressive MS is a useful approach to identify potential biomarkers and novel therapeutic targets for disease progression. Future combinations of several biomarkers may be most appropriate due to multifaceted changes of MS disease processes.

### 8.3. Chitinase-3-Like-1 Precursor

Chitinase-3-like-1 precursor (CHI3L1) has been found to be increased in the CSF of many patients with CNS inflammatory diseases and is expressed by astroglia, normal-appearing white matter, white matter plaques, and brain lesions in MS patients. Specifically, serum and CSF levels were found to be increased with the disease stage and associated with more rapid conversion to RRMS in CIS patients. Additionally, lower CSF levels were found in patients with progressive MS compared to patients with RRMS [102]. However, another study found that plasma levels of CH13L1 were increased in patients with progressive MS compared to patients with RRMS and healthy controls. Higher plasma levels were also associated with more relapses and T1 and T2-weighted lesion load and brain parenchyma fraction in patients with MS [103]. Higher CSF levels have also been associated with quicker development of disability and conversion into clinically defined MS (CDMS) in CIS patients [104]. Serum levels of CHI3L1 have been found to be increased in groups of patients unresponsive to IFN-*β* treatment [105].

### 8.4. Heat Shock Protein 70 and 90

Heat shock proteins (HSPs) are molecular chaperones, subdivided by molecular weight, that help regulate homeostasis in the CNS [106]. HSP70, which is located in the cytosol, is involved in the immune response by protecting against damage from stress in both the cell membrane and intracellular space [107,108]. In MS, it can protect neurons and oligodendrocytes from apoptosis during inflammation, but extracellular HSP70 may also play a role in inducing an immune response [109,110]. One study found that the expression of HSPA1L gene that encodes for HSP70-hom protein was correlated with an increased risk of MS development. Increased expression of HSP70-hom protein was also correlated with disease severity [111]. Another study found that MS patients had higher serum levels of HSP70 compared to healthy donors but lower levels than other inflammatory neurological diseases. The same study also found HSP70 levels to be higher in CIS and RRMS compared to PPMS or SPMS [112]. In terms of treatment response, HSP90, which produces anti-inflammatory cytokines and regulates toll-like receptor (TLR) 2 and 4 responses [113], has been shown to be more increased in the glucocorticoid receptor complex of patients that are steroid-resistance than those that are steroid-sensitive [114].

### 8.5. Kappa Free Light Chain

Kappa free light chains (KFLC) are produced during the synthesis of antibodies by plasma cells [20]. CSF Kappa free light chains have been proposed as an additional marker to aid in the diagnosis of MS with comparable sensitivity and specificity to oligoclonal bands, as they do not require a paired serum sample and provide a quick machine-operated value instead of being reliant on visual evaluation [115]. Specifically, kappa free light chain has been found to be increased in the CSF and serum of MS patients [116] and correlated with disabilities in the future [117] and disease progression as CIS patients with higher CSF levels of KFLC had an earlier conversion to clinical defined MS [118].

### 8.6. Human Endogenous Retroviruses

Human Endogenous Retroviruses (HERVs), which comprise about 8% of the human genome, are usually dormant within the genome until they are triggered by an environmental factor. Their activation can result in the production of envelope proteins by HERV-W, which appears to be involved in the pathophysiology of MS [119]. One study found that the presence of the pol gene of a MS-associated retrovirus (MSRV), such as HERV-W, in the CSF of early MS patients could point to a poorer prognosis. Specifically, they found that although MS patients that were MSRV+ or MSRV− in the CSF at study entry had similar EDSS scores, the scores were significantly different after six years. Patients in the MSRV+ were also found to have a higher annual relapse rate, along with two patients from this group having developed the progressive form of MS, while none from the MSRV− group did [120]. Therefore, more research on HERVs can prove to be beneficial as HERVs may play a significant role in understanding the development of MS and may be targets for new therapeutic remedies.

### 8.7. Uric Acid

Serum levels of uric acid, which has antioxidant properties, have been found to be decreased in patients with MS. To elucidate whether this is due to patients being primarily deficient or due to uric acid’s peroxynitrite scavenging activity, one study measured the serum urate levels of MS patients and those with other neurological diseases. They found that the urate levels of MS patients were significantly lower than those with other neurological diseases. However, no significant correlation was found between urate levels and disease activity, duration, disability, or course, supporting the notion that urate levels are primarily deficient in MS, resulting in the loss of protective effects against oxidative agents [121]. One study using a two-sample Mendelian randomization in a genome-wide association meta-analysis with 25 independent genetic variants strongly associated with serum urate levels found that increased serum urate levels do not lead to an increased risk of MS [122]. A summary of this biomarker and the ones discussed above in this section can be found in Table 5.

## 9. Biomarkers for a Future Bioinformatic Approach

### 9.1. Proteomic Approach

It may prove to be useful to include sNfL levels in a panel of other biomarkers, as discussed above, to help with prognosis in MS. Using a multiplex proteomic assay Olink technology on 724 serum proteins, a panel of serum proteins including urokinase-type plasminogen activator (uPA), kallikreins family of protease hK8, desmoglein-3 (DSG3), along with NfL were identified to be more accurate in defining a relapse than using NfL alone [123].

### 9.2. Cellular Studies

MS disease state and disease-modifying treatments (DMTs) were shown to affect the adaptive immune system, particularly T and B cell subsets, through immunological studies in MS patients [124]. The overall goal of all DMTs is to dampen the pro-inflammatory response while boosting the anti-inflammatory response by enhancing Th2, Treg, and regulatory B (Breg) cells [125]. DMTs were shown to increase Th2 and Treg cell populations while decreasing the Th1 and Th17 response. More specifically, fumarate, such as dimethyl fumarate (DMF), affected clusters of differentiation CD8+ more than CD4+, with a larger reduction seen for effector memory T (Tem) cells and central memory T (Tcm) cells than naïve T (Tn) cells [6,56,126], while sphingosine-1-phosphate (S1P) modulators affected CD4+ more than CD8+, with a larger reduction seen for Tn and Tcm than Tem. Additionally, the anti-CD20 monoclonal antibodies specifically depleted B cells [127].

Additionally, the expression and signaling of nuclear factor kappa beta (NFkB), a transcription factor involved in the regulation of the innate and adaptive immune system, has been found to be correlated with relapses in MS [128,129] and to be different between patients with RRMS and progressive MS (PMS). One study analyzing CD3+ T cells from RRMS patients discovered that out of 43 differentially expressed genes between acute relapse and complete remission, abnormal NFkB gene expression in T cells correlated most significantly with MS relapse [128]. DMTs and corticosteroids, both mainstay MS therapies, have been shown to block NFkB signaling. One study found that after methylprednisolone pulse therapy, patients with MS had significantly lower levels of DNA-binding p65 NFkB subunits than healthy controls. This demonstrates that corticosteroids result in a lower level of transcriptionally active pro-inflammatory NFkB in MS patients [130]. Another study also found that DMF treatment in MS patients decreased p65 transcriptional activity in NFkB signaling via a decrease in phosphorylation and nuclear translocation. DMF can also suppress extracellular signal-regulated kinase 1 and 2 (ERK1/2) and mitogen stress-activated kinase 1 (MSK1), which has been shown to further decrease NFkB signaling. By inhibiting these pathways, DMF can inhibit dendritic cell maturation and the differentiation of T cells into Th1 and Th17 subtypes [131].

### 9.3. Transcriptomic Approach

Future technology with single-cell RNA sequencing (sc-RNA seq) may allow researchers to compare the heterogeneity of RNA transcriptomes of individual cells within a population in addition to information about gene expression to discover future biomarkers for MS [132]. For example, using sc-RNA seq in MS, researchers have been able to further examine CSF and blood leukocytes using single-cell transcriptomics because there is increased transcriptional diversity in both specimen types. They found an increase in CSF follicular T cells that may be promoting the expansion and infiltration of B cells into the CNS in animal models, increasing the severity of MS, whereas no significant differences in cell composition were found in the blood compared to controls [133]. Another study found an increase in CSF polyclonal IgM and IgG1 B cells polarized towards an inflammatory, memory, and plasma cell phenotype, along with no detection of EBV [134].

### 9.4. Micro-RNA Molecules

Non-coding single-stranded micro-RNA (miRNA) molecules can be measured in the CSF and serum of MS patients via polymerase chain reaction (PCR) techniques. miRNAs are dysregulated in the immune system and CNS of MS patients, meaning different miRNAs can be either upregulated or downregulated in MS patients, altering gene expression of various mRNA transcripts [135,136]. Specifically, in peripheral blood mononuclear cells (PBMCs) and white matter lesions in the brain of MS patients, miR-19a, miR-21, miR-22, miR-142-3p, miR-146a, miR-146b, miR-155, miR-210, miR-233, and miR-326 are upregulated. miR-15a, miR-19a, miR-22, miR-210, and miR-223 are additionally upregulated in regulatory T cells (Tregs) and in the plasma and blood cells of MS patients. However, miR-15a, miR-15b, miR-181c, and miR-328 are downregulated [137]. One study found that there is an upregulation of serum miRNAs that promote anti-inflammation and pro-regenerative polarization in MS patients. In contrast, they found that miR-155, which promotes pro-inflammatory states, was downregulated in both primary progressive MS (PPMS) and RRMS. These findings are potentially explained by monocytes attempting to counteract the inflammation in the CNS. However, in some MS patients, especially those with progressive MS, there is a downregulation of miR-124, which promotes an anti-inflammatory state. This implies that in progressive MS, there is an absence of homeostatic monocyte control. This study also found that miR-23a, miR-30c, miR-125a, miR-146a, and miR-223 were upregulated in both RRMS and PPMS patients, but that miR-181a was only increased in RRMS [138]. Studies have also found that CSF levels of miR-181c [139] and miR-150 [140] are associated with an earlier conversion of CIS to MS. Additionally, miR-150 has been found to be upregulated in MS patients compared to controls [140].

### 9.5. Extracellular Vesicles

Extracellular vesicles (EVs) can be broken down into microvesicles and exosomes based on size. Microvesicles are usually 100–1000 nm, and exosomes are 50–150 nm. Exosomes can be used to communicate between cells and travel large distances in the body. Thus, they can be used to monitor MS disease progression and activity as well as therapeutic treatment [141]. Exosomes can be released from T cells to regulate antigen presenting cells via miRNAs contained within the exosome [142] and act as proinflammatory regulators in rheumatoid arthritis, Grave’s disease, and in MS [143,144,145]. EVs are found to be increased in the CSF and plasma of MS patients and have different molecular compositions compared to EVs from healthy individuals [141]. A study found that patients with RRMS had significantly altered miRNAs compared to controls, specifically an increase in miRNA let-7i in exosomes [146]. A higher number of total exosomes from the CSF has been described for MS patients [147]. Another study found higher CSF levels of EVs in patients with progressive forms of MS and in those with CIS. During relapses, there is an increase in the number of EVs from the CSF but was associated with decreased number of CD19+/CD200+ EVs. In addition, the presence of MS lesions was correlated with an increase of CSF EVs that were CD4+/CCR3+, CD4+/CCR5+, or CCR3+/CCR5+ [148]. During acute exacerbations of MS, there is a release of microparticles of less than 1500 nm from endothelial cells that express CD31, demonstrating endothelial dysfunction [149]. Higher levels of exosomes that express MOG were present in patients with SPMS and in relapse of RRMS patients; higher levels of MOG expression in exosomes also correlated with disease activity [150]. Additionally, exosomes from the plasma of MS patients and controls demonstrated a higher amount of C16:0 sulfatide in those from MS patients [151]. EVs from MS patients have also been found to have lower levels of TLR3 and higher levels of TLR4 compared to controls [152]. Kallikrein B1 (KLKB1) and apolipoprotein-E4 (ApoE4) were also found to be increased in the EVs of CSF compared to the CSF [153]. Acid-sphingomyelinase-enriched exosomes have also been found to be correlated with disease severity [147].

### 9.6. Metabolomics

Studying metabolites in biofluids during disease states is emerging as a powerful approach, as distinct metabolite signature could be a potential biomarker for disease progression or predictive of the beneficial effect of DMTs in MS [154]. Few studies have outlined a distinct metabolic signature, including serum phospholipids [155], altered bile acid metabolism [156], abnormalities in aromatic amino acid metabolism [157], and pro-resolving lipid mediators in MS compared to healthy subjects, which could be developed as a biomarker for disease and/or novel therapy. Moreover, altered metabolite signature during disease relapse [158,159] could be developed as a metabolic biomarker for disease progression in MS.

### 9.7. Metabolites and Gut Microbiome

With only a 30% concordance rate between monozygotic twins for MS, autoimmune demyelination is a result of both genetics and the environment. An individual must have a genetic susceptibility for MS and particular environmental factors that affect gene expression. A potential important environmental factor is an individual’s microbiome [160]. Studies have found that intestinal microbiota may affect the brain’s physiology and behaviors as well as the peripheral [161] and CNS [162] immune compartments. Stool samples provide readouts of the gut microbiota and may be useful in predicting the risk of relapse in MS [163,164] because the intestinal microbiota can affect the permeability of the BBB [165] and demyelination [166]. The microbiome of MS patients has been found to have higher amounts of Pseudomonas, Mycoplama, Haemophilus, Blautia, and Dorea genera, while the control group has higher amounts of Parabacteroides, Adlercreutzia, and Prevotella genera [163]. Another study found that MS patients had higher levels of Saccharomyces and Aspergillus, with the former being positively correlated with circulating basophils but negatively correlated with regulatory B cells, and the latter being positively correlated with activated CD16+ dendritic cells [167].

Pediatric MS patients without the gut phylum Fusobacteria have been found to have a higher risk of relapse than those with Fusobacteria [168]. MS patients have also been found to have decreased clostridial species in the gut microbiome, but they were not the spore-forming clostridial species that induce Tregs to prevent autoimmunity, demonstrating differences in clostridial species between MS and other autoimmune conditions [169].

The gut microbiota observed in MS patients also seemed to have decreased butyrate producers [169]. Studies have shown that butyrate can increase Treg populations through a short-chain fatty acid (SCFA) G-protein-coupled receptor and increase the production of anti-inflammatory cytokines, such as IL-10 and IL-4, promoting an anti-inflammatory state through IL-10 mediated activity of antigen-presenting cells (APCs) and T cells [170]. SCFAs were also found to be decreased in SPMS [170], so it may be a useful marker for progressive disease. Although links have been drawn between the microbiome and other neurodegenerative disorders [171], more studies are needed to determine if microbiota signatures can distinguish MS from other diseases and whether there is a causal relationship with microbiome disease activity in MS patients. A summary of this biomarker and the ones discussed above in this section can be found in Table 6.

## 10. Conclusions

With the search for biomarkers with more prognostic value in detecting MS relapse and progression, there have been exciting advances with NfL. However, NfL levels can be difficult to use when clinically evaluating individual patients, especially when monitoring relapse or disease progression. In addition to the many confounding variables, such as age, body mass index (BMI), and blood volume, NfL indicates neuronal damage and, thus, is nonspecific to MS. Elevated NfL also does not distinguish between patients with MS and those with minor head trauma, infection, other neurological diseases, or comorbidities, such as diabetes. Additionally, when patients have a sudden spike in their NfL levels, it is usually indicative of inflammation and active lesions. Hence, increases in NfL levels may be more indicative of neuroinflammation than neurodegeneration in MS. Other biomarkers of axonal damage, neuronal damage, glial dysfunction, demyelination, and inflammation described in this review are plagued by similar issues as NfL and are limited by conflicting results from various studies. Therefore, a combination of diverse biomarkers (protein, immune cells, transcriptomics, extracellular vesicles, metabolites, microbiome, etc.) coupled with state-of-the-art bioinformatics are needed to develop useful biomarker tools to predict true relapse and disease progression for MS patients (Figure 1). New technology like proteomics, metabolomics, and sc-RNA seq may greatly aid in the discovery of novel biomarkers and therapeutic targets for disease progression in MS.

## Figures and Tables

**Figure 1 ijms-23-05877-f001:**
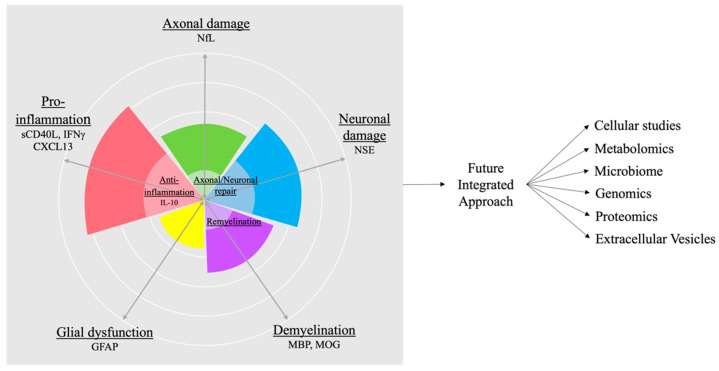
Current and future biomarkers. Currently, there are candidate biomarkers representing various processes such as demyelination, glial dysfunction, axonal and neuronal damage, as well as pro-inflammation. These may be countered by axonal and neuronal repair, remyelination, and anti-inflammation. A composite biomarker approach will help to quantify each patient disease state. A future integrated approach with bioinformatics and machine learning, combining cellular studies, metabolomics, microbiome, genomics, proteomics, and extracellular vesicles, will lead to a better understanding of each individual patient’s disease state. Better diagnostic and prognostic biomarkers will lead to better therapeutic targets and personalized therapies in the future. sCD40L = soluble CD40L; IFN = interferon; CXCL = C-X-C motif chemokine ligand.

**Table 1 ijms-23-05877-t001:** Biomarkers of Axonal Damage.

Potential Biomarker	Study Population (n)	Sample	Results	Possible Utility
NfL	RRMS (65), SPMS (10), PPMS (20)	CSF	Correlated with RRMS progression to SPMS [42]	Predictive, prognostic, treatment response
	RRMS (41), SPMS (25), controls (50)	CSF	Increased during active and acute relapse in MS patients compared to healthy controls [43]	
	RRMS (62), SPMS (3) PPMS (16), CIS (48), RIS (13), controls (87)	CSF, serum	Strong associated between CSF and serum levels; serum levels lower with disease-modifying treatment; serum levels positively correlated with age and higher in older patients during relapse and associated with higher risk of relapse and EDSS worsening [45]	
	RRMS (435), SPMS (54), PPMS (25), CIS (93)	Serum	Lower levels associated with active treatment, with larger decreases in NfL levels with high-potency treatments. Associated with T2 lesion volume over time; no association between higher levels at disease onset and higher long-term EDSS scores nor any association with relapse activity overtime; large overlap between the baseline level in MS patients and controls who may have migraine or conversion disorder [47]	
	RRMS (15)	Serum	Associated with clinical or MRI disease activity [48]	
	SET cohort: RRMS (163) GeneMSA cohort: RRMS (179)	Serum	Lower levels associated with lower probability of recent imaging disease activity; higher levels associated with higher number of active MRI lesions [49]	
	RRMS (35), PPMS (17), CIS (15)	Serum	Higher baseline levels associated with higher hazard ratio of developing EDSS ≥ 4 after 15+ years [51]	
	MS (60)	Serum	Levels were increased six years prior to onset of MS [52]	
	MS (955)	Serum	Levels were elevated only after EBV seroconversion [53]	
Tau	MS (25), controls (67)	CSF	Correlated with prominence of clinical symptoms [59]	Predictive, prognostic
	Probable or confirmed RRMS (32)	CSF	Correlated with quicker disease progression and predicts time of next relapse [60]	
	CIS (21), controls (20)	CSF, serum	No difference between CIS patients and controls; no correlation with EDSS scores [62]	
	RRMS (38), CIS (52), controls (25)	CSF	Correlated with EDSS in both CIS and RRMS patients; higher correlated with conversion of CIS into clinically defined MS; associated with the number of T2-lesions on MRI [63]	
	RRMS (32), SPMS (2), PPMS (4), CIS (12), controls (19)	CSF	Similar levels among all clinical sub-groups and controls [64]	
	CIS (20), CDMS (43), controls (56)	CSF	Similar concentrations between those with demyelinating disease and controls [68]	
APP	MS (6), controls (6)	CSF	Higher in MS patients compared to controls; MS patients with axons that are positive for APP are correlated with CNS lesion development [66]	Associated marker
TUB*β*	RRMS (24), SPMS (7), PRMS (1), PPMS (1)	CSF	Higher in MS patients than patients with other neurological diseases [67]	Associated marker

n = sample size; NfL = neurofilament light chain; APP = amyloid precursor protein; TUB*β* = tubulin *beta*; MS = multiple sclerosis; RRMS = relapsing-remitting multiple sclerosis; SPMS = secondary progressive multiple sclerosis; PPMS = primary progressive multiple sclerosis; CIS = clinically isolated syndrome; RIS = radiologically isolated syndrome; CDMS = clinically defined MS; PRMS = progressive relapsing MS; CSF = cerebrospinal fluid; EDSS = Expanded Disability Status Scale; MRI = magnetic resonance imaging; EBV = Epstein-Barr Virus; CNS = central nervous system.

**Table 2 ijms-23-05877-t002:** Biomarkers of Neuronal Damage.

Potential Biomarker	Study Population (n)	Sample	Results	Possible Utility
14-3-3	CIS (21), controls (20)	CSF	Levels are undetectable in the majority [62]	Prognostic
	CIS (20), CDMS (43), controls (56)	CSF	Associated with greater disease disability and rate of disease progression [68]	
	RRMS (10), SPMS (7), PPMS (2), controls (5)	CSF	Associated with more severe disability and extensive involvement of spinal cord [69]	
	CIS (38)	CSF	Associated with quicker progression to MS and predictive of EDSS ≥ 2 [70]	
	MS (22)	CSF	Levels are undetectable in the large majority [71]	
NSE	RRMS (41), SPMS (25), controls (50)	CSF	No difference between MS patients and controls [43]	Prognostic
	CIS (21), controls (20)	CSF, serum	Lower in CIS patients compared to controls [62]	
	RRMS (19), SPMS or PPMS (2)	Serum	Normal range in patients with MS [74]	
	RRMS (25), SPMS (23), PPMS (16)	Plasma	Negative correlated with EDSS and MSSS score [75]	

NSE = neuron specific enolase; MSSS = Multiple Sclerosis Severity Score.

**Table 3 ijms-23-05877-t003:** Biomarkers of Glial Dysfunction.

Potential Biomarker	Study Population (n)	Sample	Results	Possible Utility
GFAP	MS (503), controls (252)	CSF	Patients with SPMS had higher levels than those with RRMS [78]	Prognostic
	RRMS (20), SPMS (21), PPMS (10), controls (51)	CSF	Associated with greater disabilities and relapse [79]	
S100*β*	RRMS (41), SPMS (25), controls (50)	CSF	No difference between MS patients and controls [43]	Prognostic
	CIS (21), controls (20)	CSF, serum	No difference between CIS patients and controls; no correlation with EDSS score [62]	
	RRMS (25), SPMS (23), PPMS (16)	Plasma	No difference between various clinical subtypes of MS [75]	
	RRMS (20), SPMS (21), PPMS (10), controls (51)	CSF	Highest levels in order of PPMS, SPMS, then RRMS, with all higher than controls [79]	
	RRMS (9 with acute exacerbations, 3 stable), chronic progressive (8 with acute exacerbations, 3 stable)	Plasma	Acute exacerbations results in higher levels [80]	
AQP4	MS (144), NMO (37)	Serum	Only detectable in 4 out of 144 MS patients but detectable in 21 out of 37 NMO patients [83]	Diagnostic
	RRMS (27), SPMS (6), PPMS (5), controls (14), NMO (24)	Serum	Undetectable in all MS patients and controls, but detectable in 14 out of 24 patients with NMO [84]	
NO	RRMS (8), SPMS (8), PPMS (1), controls (8)	CSF, serum	Increased in MS patients compared to controls [85]	Associated marker
	MS exacerbation (24), MS remission (17), MS progression (20), tension headache (8), controls (11)	CSF	Increased in MS patients compared to controls [86]	

GFAP = glial fibrillary acidic protein; AQP4 = anti-aquaporin 4; NO = nitric oxide; NMO = neuromyelitis optica.

**Table 4 ijms-23-05877-t004:** Biomarkers of Myelin Biology/Demyelination.

Potential Biomarker	Study Population (n)	Sample	Results	Possible Utility
MBP	RRMS (31), CIS (18)	CSF	Correlated with EDSS scores [89]	Prognostic
	Acute exacerbation of MS (15), remission (19), slow progressive form (13)	CSF	MS patients with an acute exacerbation had higher levels than those with slower progressive MS and even higher than those in remission [90]	
MOG	RRMS (2), anti-MOG (1)	CSF	Distinct myeloid cell types if subjects with neuroinflammation [93]	Diagnostic

MBP = myelin basic protein; MOG = myelin oligodendrocyte glycoprotein.

**Table 5 ijms-23-05877-t005:** Biomarkers of Immunomodulation and Inflammation.

Potential Biomarker	Study Population (n)	Sample	Results	Possible Utility
Cytokines	RRMS (114), CIS (43)	Serum	Immunologically distinct subgroups of MS, and these subgroups may stratify treatment response to IFN-β [6]	Predictive, prognostic, treatment response
	Pediatric onset MS (40), controls (11)	Serum	IL-10 is predictive of relapse [94]	
	RRMS (323), SPMS (40), PPMS (24), CIS (79), OIND (176), ONIND (181), controls (14)	CSF	CXCL13 has been found to be correlated with worse prognosis and exacerbations in RRMS and conversion of CIS to MS [95]	
	MS (136), OND (35), controls (49)	CSF, plasma	Plasma and CSF levels of CCL11 is associated with disease duration, especially in patients with SPMS; CCL20 is associated with disease severity and CSF levels of IL-12B, MIP-1a, CD5, and CXCL9, and plasma levels of OSM and HGF to be associated with MS [96]	
	RRMS (39), controls (39)	Serum	IL-6 has been found to be correlated with age of onset for MS patients and is detected at a higher rate in MS patients compared to controls [97]	
sCD40L	RRMS (8), SPMS (32), BMS (12), controls (5)	Plasma	Significantly elevated in SPMS compared to BMS and RRMS; MCP1/CCL2 and sCD40L can be used together to differentiate between RRMS and SPMS; IFN-γ and sCD40L can be used together to differentiate between BMS and SPMS [99]	Prognostic
CHI3L1	RRMS (38), progressive MS (16), CIS (40), controls (29)	CSF, serum	Strong expression in MS patients, especially astrocytes and microglia in white matter plaques. Increased with disease stage and associated with more rapid conversion to RRMS in CIS patients. Lower CSF levels in progressive MS compared to RRMS [102]	Predictive, prognostic, treatment response
	RRMS (124), SPMS (30), PPMS (66), controls (57)	Plasma	Increased in patients with progressive MS compared to patients with RRMS and healthy controls; higher levels were associated with more relapses and T1 and T2-weighted lesion load and brain parenchyma fraction in patients with MS [103]	
	CIS (84)	CSF	Higher levels associated with quicker disease conversion to clinically defined MS in CIS patients [104]	
	RRMS (117)	Serum	Increased in groups of patients unresponsive to IFN-*β* treatment [105]	
HSP	MS (191), controls (365)	Whole blood	Expression of HSPA1L gene that encodes for HSP70-hom protein was correlated with increased risk of MS development; increased expression of HSP70-hom protein was correlated with disease severity [111]	Prognostic, treatment response
	RRMS (40), SPMS (19), PPMS (9), CIS (26), OIND (28), ONIND (41), controls (114)	Serum	Higher HSP70 levels in MS compared to healthy controls but lower than other inflammatory neurological diseases; Increased HSP70 levels in CIS and RRMS compared to PPMS or SPMS [112]	
	Steroid-resistant MS (15), steroid-sensitive MS (15)	Peripheral blood	Increased HSP90 in the glucocorticoid receptor complex of patients that are steroid-resistance compared to those that are steroid-sensitive [114]	
KFLC	RRMS (37), PPMS (4), OND (368)	CSF, serum	Increased in MS patients [116]	Predictive, prognostic
	RRMS (23), SPMS (28), PPMS (6)	CSF	Correlated with future disability [117]	
	CIS (78), controls (25)	CSF	CIS patients with higher CSF levels of KFLC had earlier conversion to clinical defined MS [118]	
HERVs	MSRV+ MS (10), MSRV- MS (8)	CSF	MSRV+ MS patients had higher EDSS scores compared to MSRV- MS patients at 6-year follow-up. MSRV+ MS patients have a higher annual relapse rate. Two patients in the MSRV+ group developed the progressive form of MS [120].	Prognostic
Uric Acid	MS (124), OND (124)	Serum	Uric acid levels are decreased in MS patients compared to those with other neurological diseases. No correlation was found between urate levels and disease activity, duration, disability, or course [121]	Associated marker
	MS (61,667), controls (86,806)	Serum	Increased urate levels do not lead to an increased risk of developing MS [122]	

CHI3L1 = Chitinase-3-Like-1 Precursor; HSP = heat shock protein; KFLC = kappa free light chain; HERVs = human endogenous retroviruses; OND = other neurological diseases; OIND = other inflammatory neurological diseases; ONIND = other non-inflammatory neurological diseases; BMS = benign multiple sclerosis; MSRV = MS-associated retrovirus; IL = interleukin; CCL = C-C motif chemokine ligand; MIP = macrophage inflammatory protein; CD = cluster of differentiation; OSM = oncostatin; HGF = hepatocyte growth factor; MCP = monocyte chemoattractant protein.

**Table 6 ijms-23-05877-t006:** Biomarkers of a Future Bioinformatics Approach.

Potential Biomarker	Study Population (n)	Sample	Results	Possible Utility
Cellular Studies	RRMS (65)	Peripheral blood	DMF shifts the balance between Th1/Th17 and Th2 and reduces memory T cells in MS patients, specifically decreasing the absolute number of CD4+ and CD8+ T cells, while increasing the CD4+/CD8+ ratio [56]	Prognostic, treatment response
	RRMS (36), SPMS (20), PPMS (43), controls (45)	Whole blood	T cell dysregulation in patients with untreated MS [124]	
	SPMS (36)	Whole blood	Siponimod treatment resulted in a decrease in CD4+ T cells, CD8+ T cells but an increase in Tem cells, Th2 cells, Tregs, and Bregs; affected CD4+ more than CD8+, with a larger reduction seen for Tn and Tcm than Tem [127]	
	RRMS (6)	Peripheral blood	Abnormal NFkB gene expression in T cells, out of 43 differentially expressed genes between acute relapse and complete remission, correlated most significantly with MS relapse [128]	
	RRMS (5), SPMS (10), PPMS (5), controls (24)	Peripheral blood	After methylprednisolone pulse therapy, MS patients had significantly lower levels of DNA-binding p65 NFkB subunits compared to that of healthy controls [130]	
Transcriptomics	MS (39), controls (27)	CSF	Follicular T cells may drive B cell expansion and infiltration in MS [133]	Prognostic
	RRMS (16), CIS (2), controls (3)	CSF	Polyclonal IgM and IgG1 B cells are polarized towards an inflammatory, memory, and plasma cell phenotype [134]	
miRNAs	RRMS (21), PPMS (8)	Serum	An overall upregulation of miRNAs that promote anti-inflammation and pro-regenerative polarization in MS patients; miR-155 is downregulated in both PPMS and RRMS and miR-124 downregulated in PPMS; miR-23a, miR-30c, miR-125a, miR-146a, and miR-223 are upregulated in both RRMS and PPMS, but that miR-181a was only increased in RRMS [138]	Predictive, prognostic
	CIS (58)	CSF	miR-181c is associated with earlier conversion of CIS to RRMS [139]	
	Cohort 1: RRMS (43), CIS (34), controls (65) Cohort 2: RRMS (96), CIS (120), controls (214)	CSF	miR-150 has been found to be upregulated in MS patients compared to controls; miR-150 is associated with earlier conversion of CIS to MS [140]	
EVs	RRMS (4), controls (4)	Plasma	An increase in miRNA let-7i in the exosomes of MS patients [146]	Prognostic
	RRMS (21), OND (20)	CSF	Higher number of total exosomes in MS patients; ASM-enriched exosomes correlated with disease severity [147]	
	RRMS (35), progressive MS (4), CIS (2), OIND (2), ONIND (16)	CSF	Higher levels of EVs in patients with CIS and progressive forms of MS; increase in the number of EVs during relapse but decrease in number of CD19+/CD200+ EVs; presence of MS lesions is correlated with an increase of CSF EVs that were CD+/CCR3+, CD4+/CCR5+, or CCR3+/CCR5+ [148]	
	MS exacerbation (30), MS remission (20), controls (48)	Plasma	Release of microparticles of less than 1500 nm from endothelial cells that express CD31 during acute exacerbations [149]	
	RRMS (45), SPMS (30), controls (45)	Serum	Higher levels of exosomes that express MOG were present in patients with SPMS and in relapse of RRMS patients; higher levels of MOG expression in exosomes also correlated with disease activity [150]	
	RRMS (8), SPMS (1), controls (9)	Plasma	Exosomes from MS patients have increased C16:0 sulfatides compared to controls [151]	
	RRMS (18), controls (16)	Serum	EVs from MS patients have lower levels of TLR3 but higher levels of TLR4 compared to controls [152]	
	RRMS (4), controls (3)	CSF	KLKB1 and ApoE4 are increased in EVs of CSF compared to the CSF [153]	
Metabolomics	RRMS (24), controls (30)	Plasma	Decreased levels of PC(34:3), PC(36:6), PE(40:10) and PC(38:1) phospholipids [155]	Prognostic
	RRMS (106), PMS (176), controls (127), pediatric MS (31), pediatric controls (31)	Plasma	Decreased secondary bile acids [156]	
	MS (637), controls (317)	Plasma	Alteration in aromatic amino acid metabotoxins [157]	
	Retrospective longitudinal cohort: MS (238), controls (74) Prospective cohort: MS (61), controls (41)	Plasma	Identified metabolic signature consist of hormones, lipids, and amino acids associated with MS and with a severe disease course [158]	
	RRMS in relapse (38), last relapse (LR) between 1 to 6 months (28), LR between 6–24 months (34); LR more than 24 months ago (101)	Plasma	Identified four metabolites including lysine, asparagine, isoleucine, and leucine, which showed a consistent trend with time away from relapse [159]	
Metabolites and microbiome	RRMS (31), controls (36)	Microbiome	MS patients had higher amounts of Pseudomonas, Mycoplama, Haemophilus, Blautia, and Dorea genera, while the control group had higher amounts of Parabacteroides, Adlercreutzia, and Prevotella genera [163]	Prognostic
	RRMS (21), SPMS (1), PPMS (2), controls (22)	Microbiome	MS patients had higher levels of Saccharomyces and Aspergillus, with the former being positively correlated with circulating basophils but negatively correlated with regulatory B cells, and the latter positively correlated with activated CD16+ dendritic cells [167]	
	Pediatric RRMS (17)	Microbiome	Absence of Fusobacteria is associated with quicker relapse compared to the presence of Fusobacteria [168]	
	RRMS (20) controls (58)	Microbiome	Decreased cloistral species and butyrate producers in MS patients [169]	
	SPMS (20), controls (15)	Plasma	SCFAs were also found to be decreased in SPMS [170]	

miRNAs = micro-RNAs, EVs = extracellular vesicles; DMF = dimethyl fumarate; NFkB = nuclear factor kappa beta; Th = T helper; Tem = effector memory T cells; Tregs = regulatory T cells; Bregs = regulatory B cells; Tn = naïve T cells; Tcm = central memory T cells; Ig = immunoglobulin; ASM = acid sphingomyelinase; TLR = toll-like receptor; KLKB1 = kallikrein B1; ApoE4 = apolipoprotein-E4; PC = phosphatidylcholine; PE = phosphatidylethanolamine; SCFAs = short-chain fatty acids.

## Data Availability

Not applicable.

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
