# Peer review of "Current and Future Biomarkers in Multiple Sclerosis"

_ijms, 2022, doi:10.3390/ijms23115877_

Round 1

Reviewer 1 Report

To the authors:

Overall, the subject is very interesting: standardization of diagnosis and prognosis of multiple sclerosis is of great value in clinical practice. Accurate determination of relapses in multiple sclerosis is important for diagnosis, classification of clinical course and therapeutic decision making. The authors provide a concise overview of markers for the diagnosis and prognosis of multiple sclerosis. I have only two minor issues with the work presented, but am otherwise very enthusiastic about the presentation, flow and coverage of the manuscript:

Minor issues

i) Page 24, lines 912-913: reference 118 seems unpublished. I suggest the use of bioRxiv or similar services for or unpublished preprints in the future.

ii) Page 14, lines 453-465: The assessment of cellular phenotypes and molecular traits and translation to clinical use is currently a fascinating area of active research, with NFkB pathway family showing some promise. I suggest the authors consider adding NFkB as a biomarker. NFκB is a transcription factor and master regulator of both innate and adaptive immunity, playing critical roles in activation, proliferation, differentiation, and cytokine production in B cells, T cells, DCs, monocytes/macrophages and NK cells. It been suggested as both a potential treatment target, and a biomarker for relapses in MS (1, 2). Corticosteroids block NFκB activation and are a routine MS therapy; patient response is associated to overall decrease in NFκB activity. Several other MS disease-modifying agents, including dimethyl fumarate (DMF), also inhibit the NFκB pathway (3, 4).

1. Satoh, J.-I., Misawa, T., Tabunoki, H. & Yamamura, T. Molecular network analysis of T-cell transcriptome suggests aberrant regulation of gene expression by NF-kappaB as a biomarker for relapse of multiple sclerosis. Dis. Markers 25, 27–35 (2008).

2. Yan, J. & Greer, J. M. NF-kappa B, a potential therapeutic target for the treatment of multiple sclerosis. CNS Neurol Disord Drug Targets 7, 536–557 (2008).

3. Eggert, M. et al. Changes in the activation level of NF-kappa B in lymphocytes of MS patients during glucocorticoid pulse therapy. J. Neurol. Sci. 264, 145–150 (2008).

4. Peng, H., Guerau-de-Arellano, M. & Mehta, V. B. Dimethyl fumarate inhibits dendritic cell maturation via nuclear factor κB (NF-κB) and extracellular signal-regulated kinase 1 and 2 (ERK1/2) and mitogen stress. Journal of Biological Chemistry 287:28017-26 (2012).

Reviewer 2 Report

Yang J: Current and Future Biomarkers in Multiple Sclerosis. IJMS 

this is a vast review that can be useful for readers as it summarizes in a few pages years of research aimed at identifying diagnostic and prognostic markers in MS. 

Future perspectives raising from the application of computer mining methodologies have also been addressed. I have only a few suggestions to submit to the authors:

- line 93: " .....as anything causing inflammation .... "; better say ...."as anything causing chronic inflammation..:"

- line 152. "A study of U.S. military personnel ........ before the clinical onset of MS [52]". To this point, please consider that an extension of this study has provided new evidences that MS risk increased 32-fold after EBV infection and that serum neurofilaments increased only after EBV seroconversion (Bjornevik K, et al. Longitudinal analysis reveals high prevalence of Epstein-Barr virus associated with multiple sclerosis. Science. 2022 doi: 10.1126/science.abj8222.). Authors are required to comment on this important study by Ascherio and colleagues.

Other inflammatory/prognostic markers that could be added:

  • HERVs (Küry P, et al. Trends Mol Med. 2018 Apr;24(4):379-394. doi: 10.1016/j.molmed.2018.02.007; Sotgiu S, et al. Multiple sclerosis-associated retrovirus in early multiple sclerosis: a six-year follow-up of a Sardinian cohort. Mult Scler. 2006 Dec;12(6):698-703. doi: 10.1177/1352458506070773.
  • Uric acid: Sotgiu S, et al. Serum uric acid and multiple sclerosis. Neurol Sci. 2002 Oct;23(4):183-8. doi: 10.1007/s100720200059. Harroud A, et al. Neurol Neuroimmunol Neuroinflamm. 2020 Nov 19;8(1):e920. doi: 10.1212/NXI.0000000000000920. 
